# Energy-Efficient UAVs Deployment for QoS-Guaranteed VoWiFi Service

**DOI:** 10.3390/s20164455

**Published:** 2020-08-10

**Authors:** Vicente Mayor, Rafael Estepa, Antonio Estepa, Germán Madinabeitia

**Affiliations:** Department of Telematics Engineering, Universidad de Sevilla, 41092 Seville, Spain; rafa@trajano.us.es (R.E.); aestepa@trajano.us.es (A.E.); german@trajano.us.es (G.M.)

**Keywords:** UAV, voice over IP, wireless LAN, IEEE 802.11, voice over IP over WiFi, VoWiFi, quality of service, energy

## Abstract

This paper formulates a new problem for the optimal placement of Unmanned Aerial Vehicles (UAVs) geared towards wireless coverage provision for Voice over WiFi (VoWiFi) service to a set of ground users confined in an open area. Our objective function is constrained by coverage and by VoIP speech quality and minimizes the ratio between the number of UAVs deployed and energy efficiency in UAVs, hence providing the layout that requires fewer UAVs per hour of service. Solutions provide the number and position of UAVs to be deployed, and are found using well-known heuristic search methods such as genetic algorithms (used for the initial deployment of UAVs), or particle swarm optimization (used for the periodical update of the positions). We examine two communication services: (a) one bidirectional VoWiFi channel per user; (b) single broadcast VoWiFi channel for announcements. For these services, we study the results obtained for an increasing number of users confined in a small area of 100 m^2^ as well as in a large area of 10,000 m^2^. Results show that the drone turnover rate is related to both users’ sparsity and the number of users served by each UAV. For the unicast service, the ratio of UAVs per hour of service tends to increase with user sparsity and the power of radio communication represents 14–16% of the total UAV energy consumption depending on ground user density. In large areas, solutions tend to locate UAVs at higher altitudes seeking increased coverage, which increases energy consumption due to hovering. However, in the VoWiFi broadcast communication service, the traffic is scarce, and solutions are mostly constrained only by coverage. This results in fewer UAVs deployed, less total power consumption (between 20% and 75%), and less sensitivity to the number of served users.

## 1. Introduction

Unmanned Aerial Vehicles (UAVs) for wireless communications have experienced rapid growth in a broad range of application domains [1]. On-board communication devices enable UAVs to enhance coverage, capacity, reliability or energy efficiency of wireless networks. However, note that the drone communication technology has a decisive impact on the data rate and range usable in the mission [2]. The predominant technology in UAV-related literature is cellular (i.e., UAVs are used as aerial base stations) [3], but it is not uncommon to find studies with other communication technologies for UAV-to-ground communication such as WiFi [4,5,6] or Bluetooth [7]. This work takes WiFi as enabling technology for UAV-to-ground communication. WiFi provides a data range of 11 Mbps–1 Gbps and a range of up to 250 m (depending on the revision).

Deploying UAVs for wireless networks brings in new opportunities, such as creating a provisional communication infrastructure in Search and Rescue (SAR) missions [1,8]. However, it also brings multiple challenges such as optimal 3D placement, performance analysis, path planning, backhaul connectivity, or energy limitation, just to name a few [3,9]. In this work, we deal with the challenge of optimal multi-UAV 3D placement in the context of creating a provisional WLAN (i.e., UAVs are used as aerial Access Points—AP) to provide ground users with VoIP over WiFi (VoWiFi) service. A distinctive characteristic of this service is that 3D placement has to consider not only coverage (e.g., [10,11]) but also VoIP speech quality [12] since the mac-sublayer of the communication protocol limits the number of simultaneous VoIP flows that APs can handle with quality of experience (for example, an AP performing IEEE 802.11 b can handle a maximum of 15 simultaneous calls [13], even though the number of stations associated to such AP could be multiple times greater). This limit is known as VoIP capacity [14,15,16].

UAV’s optimal 3D placement is a challenging task that has received significant attention from the research community [3,9,17,18]. The problem has to take into account multiple factors such as the terrain, location of users, UAV-to-ground channel characteristics or radio interference between UAVs, and in our case, the Quality of Service (QoS). According to Sharkoor et at. [19], another factor to be carefully taken care of “when deploying UAVs is their flight time and energy constraints”. Indeed, the authors provide an excellent literature review with emphasis on the energy efficiency perspective, dissecting the factors affecting energy consumption in UAVs such as UAV placement, UAV-user association, flight-time, mobility, trajectory, and UAV communication and mechanical energy. Although optimal 3D placement has been extensively addressed, the comparison between works is difficult as they have different objective functions and constraints. In general, we can classify them roughly as (a) those that minimize the number of UAVs deployed [20,21,22,23] (i.e., minimize deployment cost); (b) those maximize energy efficiency or battery lifetime [24,25,26] (i.e., UAV endurance [27]). Energy-aware UAV optimal deployment problems typically considering the power consumption due to either radio communication [25,26] or mechanical energy [24] using energy consumption models [28,29]. The reader is encouraged to read the surveys [2,19,30] for a more detailed list of works and references that address the problem of 3D placement with emphasis in energy consumption.

### 1.1. Motivation

Long-duration activities may require hot replacement of drones due to battery exhaustion. In this work, we would like to deploy aerial APs to provide VoWiFi service to ground users in an outdoor long-duration activity (e.g., concert or similar longer 20–25 min) that will likely require hot replacement of drones due to battery exhaustion. In this context, the following considerations should be taken into account:
Besides coverage, the multi-UAV 3D optimal placement has to consider VoWiFi speech quality for which the mac-sublayer of the communication protocol is crucial. To the best of our knowledge, the only work that has considered this factor [12] sets the focus only in minimizing the number of UAVs deployed (among solutions with a similar number of drones, the one that extended the battery life of served users was chosen) disregarding energy efficiency in UAVs.The number of UAVs required for the mission depends not only on the UAVs initially deployed but also on the number of UAVs launched to replace those in battery exhaustion. Thus, besides minimizing the number of UAV initially deployed, we should also seek the placement that maximizes the UAVs operation lifespan, considering UAV communication and mechanical energy.Long-duration events should consider users’ mobility. So besides the initial 3D placement, there should be a method in place so that UAVs dynamically relocate themselves for service continuity.

### 1.2. Contribution of This Paper

This paper defines and solves a new optimization problem that finds the number and position of UAVs to be deployed over an area to provide VoWiFi service to a set of ground users. Our optimal 3D placement minimizes the number of UAVs deployed over time (i.e., rate of drones per hour of service) while satisfying coverage and QoS constraints.

The contributions of this paper can be summarized as follows:
We mathematically formulate a new UAV optimal deployment problem that accounts for UAVs turnover rate, coverage, and quality of service.We provide an analytical model that predicts the speech quality in a IEEE 802.11 WLAN for a set of VoIP traffic sources.We provide an analytical model that predicts UAV energy expenditure due to wireless communications. This information is used along with flight energy consumption to predict UAVs endurance.We solve the UAV optimal deployment problem using well-known heuristic methods such as genetic algorithm and particle swarm optimization.

## 2. Related Works

Multiple survey papers address the optimal placement of UAVs for wireless communications. In this section, we scrutinize four recent surveys and complement them with other related works applicable. Our goal is to clarify how our work compares with state of the art.

The authors in [2] review the literature for multiple drone applications such as drone base-stations, search and rescue, or surveillance and monitoring, which is challenging due to the disparity of goals and constraints. Regarding energy efficiency and 3D placement, in [26], the authors suggest an algorithm that minimizes the transmit power for the 3D placement of one drone and assess their approach by simulation. The 3D placement of one aerial base station was also proposed in [25]. In this case, the authors proposed a placement algorithm to maximize the number of covered users using the minimum transmit power. In both cases, only the power related to radio communication was taken into account. In [31] the authors propose a placement algorithm for a set of UAV-mounted 5G aerial base stations that maximizes the capacity boost provided by the UAVs in each considered time frame and extends the battery life of the served mobile users. Unfortunately, the authors did not model the energy consumption, stating that “intuitively, less power is needed if the Gateway is closer to the users”. In a context similar to ours, in [12], we proposed a 3D placement algorithm that minimized the number of aerial APs deployed to provide VoWiFi to ground users. To the best of our knowledge, this is the only related work in which speech quality has been considered a constraint. However, the minimization subject was the number of UAVs launched, and only among solutions with a similar number of drones, the algorithm selected the layout that extended the battery life of served users, overlooking the power expenditure in UAVs.

The survey paper [3] addresses the use of UAVs for wireless networks. The paper provides state of the art in the optimal deployment of UAVs as flying base stations. In their previous work [32], the authors addressed the optimal deployment and mobility of multiple UAVs for energy-efficient data collection, minimizing the total transmit power of IoT devices (i.e., ground users) during UAVs trajectories. Other works such as [18,22,23,33,34,35,36,37,38] address the 3D optimal placement problem considering coverage of ground users through different approaches, but disregard energy efficiency. Some of these works consider interference between UAVs [33].

Another excellent paper is [30], where the authors survey recent research addressing 1–3D placement optimization for aerial base stations. The authors highlight the diversity of objectives in the optimization problem, such as maximizing the system capacity, minimizing the required number of UAVs [34], or minimizing the total transmit power of the entire aerial system [39]. The algorithms used for the optimal placement included force brute search [37], Genetic Algorithms [40], K-means clustering [41] or transport theory [39]. This survey also provides a review of power-efficient operation of aerial base stations emphasizing that researchers typically work on either reducing communication energy (i.e., minimizing transmission power) of either the drone or ground users [17], or reducing mechanical energy of UAVs. The latter requires to count with an energy consumption model [21], which largely depends on hovering. As the authors put it, “one solution to control the energy consumption of UAVs is to regulate their height. However, changing the height might reduce the performance of UAVs”. This trade-off is also applies to this work and it will be addressed in the next section. Finally, the survey provides a list of testbeds and real experiments in the literature. Interestingly, although most UAV communication research focuses on deploying UAVs as aerial base stations (i.e., cellular technology such as 3G/4G/5G), most testbeds use WiFi (IEEE 802.11 b/g/a) as communication protocol such as in this work.

Only in [19] one can find a literature review with particular emphasis on the energy-efficiency perspective. Besides pointing to +20 other UAV communication-related existing surveys, the authors discuss the main factors affecting UAVs’ energy consumption. One of these factors is UAV placement. As such, the authors review energy-efficiency algorithms for optimal UAV placement in the literature. [42] suggests using Machine Learning for the optimal placement of UAV-mounted base stations given a predicted traffic demand. In their approach, they first find the optimal partitions of service areas. Then, the optimal location of each UAV that minimizes the total power consumption is derived. While the authors model the transmit power of UAVs deployed, they fail to use a mechanical energy model and simply minimize the UAVs travel distance, assuming that the mechanical power is proportional to it. [43] proposes a solution to minimize UAV’s mechanical and communication energies while integrating cognitive radio technology. The optimization problem is solved using a deterministic algorithm based on the Weber formulation and its performance is compared to that of a meta-heuristic algorithm, namely particle swarm optimization algorithm (PSO). In this case, the authors model the total energy consumption of a drone with two components, the hover and transmission energies.

Finally, Table 1 compares the main related works with our proposal. This work’s uniqueness lies in a new objective function for the problem of multi-UAV optimal 3D placement. The consideration of WiFi and its mac-sublayer on the speech quality (as a constraint) can only be found in [12]. However, [12] does not model the energy consumption in the UAVs and has a different objective function, as so does other works that model the mechanical and communication power in UAV-installed base stations such as [42,43].

## 3. QoS and UAVs Endurance in VoWiFi Networks

The optimization problem addressed in this work provides the number and position of a set of UAVs to the deployed over a known area to provide WoWiFi service to a set of ground users at known locations. Our objective function minimizes the ratio of drones per hour required to provide this service constrained to user’s coverage and user’s speech quality (i.e., QoS). In this section, we study the relationship between drone position, QoS, and UAV endurance.

Figure 1 illustrates two UAV-mounted APs providing VoWiFi to ground users (note that only the UAV-to-user communication is considered and the communication between UAVs or with the data network backhaul is taken for granted). On the leftmost part, a lower-altitude drone provides coverage to four users. Their traffic (red lines show VoIP traffic flows) is aggregated at the aerial AP. This traffic load may influence the QoS perceived by these users and the UAV’s power consumption due to the transmission/reception of packets. The rightmost UAV is covering eight ground users so its AP aggregates more traffic which may have an impact on the QoS perceived by ground users. Hovering at a higher altitude and servicing more users will likely result in more energy expenditure due to both hovering and more transmission/reception. Thus, the rightmost drone is expected to be replaced earlier than the leftmost drone due to earlier battery exhaustion. This simple example illustrates how the decision variables of the optimization problem (i.e., number and position of drones) have an impact on both, the level of speech quality perceived by VoIP users and UAVs’ energy expenditure. Next, we provide the analytical expressions used to model VoWiFi’s QoS and UAV’s energy expenditure as both aspects are central in the problem’s objective function.

### 3.1. Modeling QoS in VoWiFi

Each aerial AP and its associated stations can be studied as an independent IEEE 802.11 system, as illustrated in Figure 2. Although all stations have a common Medium Access Control protocol (i.e., MAC sub-layer), the Modulation and Coding Scheme (MCS) set on each station is automatically self-configured seeking, the greatest possible data bit-rate achievable with the present signal strength. Table 2 shows different MCSs defined for OFDM modulations (e.g., 802.11 a/g) and the minimum signal strength required for each one.

VoIP is a mature technology that counts with several methods for the assessment of voice quality [44]. At the planning stage, the method predominantly used in the literature for QoS assessment is the E-Model [45] since it allows one to estimate the speech quality coming down to the effect that the VoIP traffic load has on network performance in terms of packet loss and delay. The E-model provides a quality score termed *R* factor from 0 (poor) to 100 (excellent) which can be readily obtained using the following expression [12,46,47]:(1)$$\begin{array}{cc} R & =R_{0}-\underset{I_{e},\mathrm{eff}}{\underbrace{I_{e}+\left(95-I_{e}\right)\frac{P_{pl}}{\frac{P_{pl}}{\mathrm{BurstR}}+B_{pl}}}}\\ \; & -\underset{I_{d}}{\underbrace{\left(0.024d+0.11\cdot (d-177.3)\cdot H(d-177.3)\right)}} \end{array}$$
where:$I_{e},\mathrm{eff}$ represents a combination between impairment equipment parameter at zero packet loss ($I_{e}$), and a function that depends on $I_{e}$, the packet loss rate, and packet loss behaviour. $I_{e}$ is a codec-dependent constant associated with codec compression degradation (a list of values from ITU-T codecs were presented in ITU-T Rec. G.113 Appendix I), $B_{pl}$ represents the codec packet loss robustness, which also has a specific value for each codec (listed in ITU-T Rec. G117 Appendix I), $P_{pl}$ represents the packet loss rate (in %) in the WiFi channel and finally BurstR characterizes the burst ratio (i.e., equals 1 if packet loss if random and greater otherwise). In this paper, we use the G.711 codec ($I_{e}=0$ and $B_{pl}=25.1$) and assume random losses (i.e., BurstR=1).$I_{d}$ accounts for impairments associated with the delay in the communication chain. A widely accepted approximation for $I_{d}$ can be obtained from one-way delay in the communication path *d*, where *H* is the Heaviside function (H(x)=0 for x<0 and H(x)=1 for x>0. In our case x=d−177.3). The one-way delay *d* (in milliseconds) includes the delay in the WiFi network plus 20 ms for the packetization interval of the codec.

All terms in Equation (Equation 1) are known but the network delay *d* and packet loss rate $P_{pl}$. The values for these two factors will be obtained from our WiFi mac-sublayer analytical model for each UAV deployed (we assume that backhaul networks do not introduce any negative impacts into QoS (i.e., infinite bandwidth), or it can be considered as a constant in *R*) (and its associated users). *R* is one of our objective function constraints, and its computation will be addressed in Section 5.2.2.

### 3.2. Modeling UAV Endurance

Endurance can be described as the total time taken during the flight ($T_{\mathrm{Flight}}$) until the depletion of the UAV’s battery. High-power performance commercial UAVs are typically powered by LiPo batteries, whose energy density varies between 150 and 300 Wh/Kg. In this work, the UAV energy expenditure will consider two factors: flight and radio communication, being the former dominant in the energy budget.

According to [28], the energy (in Wh) associated with flying can be broken down into three different flight stages:(2)$$E_{\mathrm{Flight}}=E_{\mathrm{Climb}}+E_{\mathrm{Translational}}+\underset{E_{\mathrm{Hover}}}{\underbrace{P_{\mathrm{Hover}}\cdot T_{\mathrm{Hover}}}}$$
where $E_{\mathrm{Climb}}$ stands for the energy associated with vertical climbing, $E_{\mathrm{Translational}}$ is the energy spent in lateral transitions, and $E_{\mathrm{Hover}}$ accounts for the energy spent while hovering (i.e., static). As proved in [48], the hovering factor is dominant and overshadows the other two, and thus, the expression in Equation (Equation 2) can be simplified to its last term.

Besides Equation (Equation 2), the other contributing factor to energy consumption is the power spent on the radio used by the VoWiFi service.
(3)$$E_{\mathrm{Radio}}=E_{\mathrm{WiFi}}+E_{\mathrm{Backhaul}}$$
where $E_{\mathrm{WiFi}}$ represents the energy consumed by the wireless IEEE 802.11 card of the aerial AP, and $E_{\mathrm{Backhaul}}$ accounts for the energy consumed by the radio link used for the backhaul network.

Assuming that network cards are not running power saving mechanisms, the first term of Equation (Equation 3) ($E_{\mathrm{WiFi}}$) can be further decomposed as:(4)$$E_{\mathrm{WiFi}}=P_{\mathrm{WiFi}}\cdot T_{\mathrm{Flight}}=\rho_{\mathrm{tx}}\cdot T_{\mathrm{tx}}+\rho_{\mathrm{rx}}\cdot T_{\mathrm{rx}}+\rho_{\mathrm{idle}}\cdot T_{\mathrm{idle}}$$
where $T_{\mathrm{tx}}$, $T_{\mathrm{rx}}$ and $T_{\mathrm{idle}}$ stand for the time spent by the network interface card on transmission, reception and idle states respectively (their sum must be $T_{\mathrm{Flight}}$); and $\rho_{\mathrm{tx}}$, $\rho_{\mathrm{rx}}$ and $\rho_{\mathrm{idle}}$ stand for the respective power consumption coefficients (which depend on the network interface card). The time taken on each state depends on the traffic processed at the AP during the flight, which in turn depends on the VoIP traffic pattern, the MCS used at each station, and retransmissions that may occur due to noise or collisions. In Appendix A we derive closed-form expressions for the temporal factors in Equation (Equation 4) for the IEEE 802.11 mac sub-layer.

Regarding the second factor in Equation (Equation 3), (the backhaul radio link is out of the scope of our this work and the radio power will depend on the particular technology used. However, the assumption indicated in Equation (Equation 5) is acceptable for the sake of completeness) we can assume that in general, it will be proportional to the WiFi consumption, as expressed in Equation (Equation 5):(5)$$E_{\mathrm{Backhaul}}=\underset{P_{\mathrm{Backhaul}}}{\underbrace{K\cdot P_{\mathrm{WiFi}}}}\cdot T_{\mathrm{Flight}}$$
where *K* is a constant that represents the relation between both radio consumptions. For the remainder of this paper we use K=1 so $P_{\mathrm{Backhaul}}=P_{\mathrm{WiFi}}$.

Therefore, the total energy *E* spent by the UAV can be expressed as in Equation (Equation 6):(6)$$E=E_{\mathrm{Flight}}+\underset{E_{\mathrm{Radio}}}{\underbrace{E_{\mathrm{WiFi}}+E_{\mathrm{Backhaul}}}}$$

Finally, assuming hovering as the dominant factor in Equation (Equation 2), the UAV’s expected flight duration (in hours) can be expressed as:(7)$$T_{\mathrm{Flight}}=\frac{E_{\mathrm{Battery}}}{P_{\mathrm{Hover}}+\left(1+K\right)\cdot P_{\mathrm{WiFi}}}$$
where $E_{\mathrm{Battery}}$ represents the battery energy in Watts per hour. $T_{\mathrm{Flight}}$ is one of the factors of our objective function and its computation will be addressed in greater detail in Section 5.2.3.

## 4. Problem Statement

### 4.1. Terminology and Assumptions

As in most related works, we discretize the flying space, as illustrated in Figure 3. The collection of edges of the grid form a set of $R^{3}$ coordinates denoted as P. These edges define potential locations for UAVs. Each UAV mounts one AP. As such, the words UAV and AP may be used interchangeably for the remainder of this work.

Throughout this paper, we use the following definitions and terminology:Drones are denoted by the set D={1,2,…D}. Their location is given by the set $X=\{x_{1},x_{2},\ldots ,x_{D}|x_{i}\in P,i\in D,x_{i}\neq x_{j}{|}_{\forall j\neq i}\}$, where $x_{i}\in P$ represents the three-dimensional coordinates of drone *i*.Ground users are denoted by the set U={1,2,…,U}. Their location is given by $\left\{w_{k}|k\in U\right\}$, where $w_{k}$
$\in R^{3}$ represents the coordinates of user *k*.The subset of users (i.e., WiFi clients) associated to the AP installed at drone i∈D is denoted by C(i)⊂U. Thus, the number of ground users covered is given by $C=\sum_{i=1}^{D}\left|C(i)\right|$.R(i) represents the expected speech quality level (from 0 to 100) for users associated to the AP installed at drone i∈D.$T_{\mathrm{Flight}}\left(i\right)$ is the expected flight duration for drone *i* (see Equation (Equation 7), UAV’s endurance). The average flight duration of the set of UAVs deployed, ${\bar{T}}_{\mathrm{Flight}}$, is:
(8)$${\bar{T}}_{\mathrm{Flight}}=\frac{1}{D}\cdot \sum_{i=1}^{D}T_{\mathrm{Flight}}\left(i\right)$$

For the sake of simplicity and tractability, the following assumptions are considered:The terrain has known dimensions, and it is an open area.The number of drones available for the initial deployment has an upper bound $D_{\max}$.Ground users have a smartphone with a compatible VoIP application installed (i.e., using a known codec).User positions are known. This assumption is commonly found in literature and could be implemented through user’s smartphone’s GPS or image processing.APs’ channelization is arranged in such a manner that interferences between adjacent radio channels are negligible.

### 4.2. Problem Definition

Our deployment problem provides the optimum number and position of UAVs that are required to provide VoWiFi service to ground users. We aim to minimize the number of UAVs deployed over time (i.e., rate of drones per hour of service) while satisfying coverage and QoS constraints. For example, for a service duration of one hour, a solution with 3 drones that last 30 min (6 drones/h) will be preferred to that with 2 drones that last 15 min (8 drones/h). Therefore, our objective function minimizes the ratio between the number of UAVs deployed (*D*) and their average endurance expectation (${\bar{T}}_{\mathrm{Flight}}$). This can be formulated as follows:(9)$$\begin{array}{cc} \min_{X} & \frac{D^{2}}{\sum_{i=1}^{D}T_{\mathrm{Flight}}\left(i\right)}\\ \mathrm{subject}\;\mathrm{to} & \sum_{i=1}^{D}\left|C(i)\right|/U\geq C_{\min}\\ & R\left(i\right)\geq R_{\min}\;\;,\forall i\in D\\ & x_{i}\in P\;\;,\forall i\in D\\ & D\leq D_{\max} \end{array}$$
where *D* represents the number of UAVs deployed, $T_{\mathrm{Flight}}\left(i\right)$ accounts for drone *i* expected flight duration, $C_{\min}$ stands for the minimum ratio of users that must be covered, and $R_{\min}$ is a constant that defines the minimum acceptable speech quality (e.g., 65 as proposed in ITU-T G.107 [45]).

## 5. Solving a Pseudo-Static Scenario

For small flying spaces, solutions could be found through exhaustive search by assessing the objective function (i.e., see (Equation 9)) for every grid edge for an increasing number of UAVs, starting with one (D=1). If no valid solutions were found, the number of drones should be increased until a solution is found, or a maximum number of UAVs is reached ($D=D_{\max}$). Once a solution is found, the search method has to consider launching an extra drone to ensure that the objective function value cannot be further reduced (i.e., increasing the number of UAVs could reduce the UAV turnover rate). However, although exhaustive search is feasible in very small areas, it is not generally feasible in most real-life scenarios, for assessing a solution with *D* drones would require to evaluate $\sum_{d=1}^{D+1}\left(|P|/d\right)$ possibilities. Thus, we suggest using heuristic search algorithms as a general method to find solutions to our optimization problem. Although several other metaheuristic methods are applicable, we opt for using Genetic Algorithms to find a sub-optimal solution due to its simplicity and proven effectiveness in similar problems [12,40,49], since the contribution of this work is not focused on the performance of the metaheuristics method.

### 5.1. Search Algorithm: GA

A genetic algorithm is a heuristic search method inspired by the process of natural selection. A population of individuals (i.e., candidate solutions to the problem at hand) is evolved through evolutionary operators (e.g., crossover, mutation) toward better solutions. For every new generation, the population is evaluated and scored according to a fitness function and the best individuals are passed to the next generation. This process typically starts with a first generation of random individuals.

In our problem, each individual represents the location of *D* drones $X=\{x_{1},x_{2},\ldots ,x_{D}|x_{i}\in P,i\in D,x_{i}\neq x_{j}{|}_{\forall j\neq i}\}$, where $x_{i}\in P$ stands for the location of drone *i*. For example, consider a scenario with D=3 drones; then, individuals would be represented by the following sequence of genes:(10)$${X|}_{D=3}=\left\{\underset{x_{1}}{\underbrace{x_{1},\;y_{1},\;z_{1}}},\;\underset{x_{2}}{\underbrace{x_{2},\;y_{2},\;z_{2}}},\;\underset{x_{3}}{\underbrace{x_{3},\;y_{3},\;z_{3}}}\right\}$$

Algorithm 1 shows the pseudocode of our GA-based search method. The input of the algorithm is the set of ground users (U) and their location ($w_{k}$), the set of edges (P), the minimum coverage constraint ($C_{\min}$), the minimum speech quality constraint ($R_{\min}$), and the maximum number of available drones ($D_{\max}$). The algorithm starts by seeking solutions with one drone (D=1 ) and then increases the number of UAVs until a sub-optimal solution is found (or until $D_{\max}$). On each iteration, the genetic algorithm (lines from 3 to 18 in Algorithm 1) aims to find the best solution that meets the problem constraints. Every time that the genetic algorithm is initiated, an initial population of candidate solutions (i.e., individuals) is created as I by the *GA_InitialPopulation* function. Then, every individual in I is evaluated using the *Check* function. This function returns the objective function value (see Equation (Equation 9)) including the ratio of coverage and QoS-level. If one individual meets the problem constraints and improves the last best value of the objective function ($f_{\mathrm{obj}}^{*}$), the individual is recorded as a candidate solution. After the assessment of one generation, the next one is created. In the next generation, some individuals are a copy of the best individuals from parents (function *GA_Elite*), some others are created applying operations of Crossover-and-Mutation (function *GA_CM*) and Self-Reproduction and Mutation (function *GA_SRM*) to the parents [50,51]. This process is repeated until a maximum number of generations is reached without any improvement (exit criteria).
**Algorithm 1:** GA-based search pseudocode.
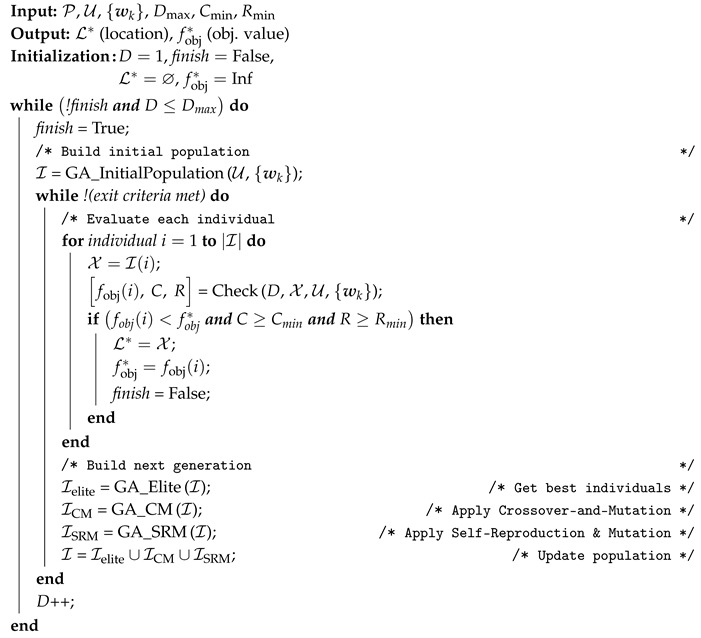


The remainder of this section, elaborates on the assessment of individuals (i.e., *Check* function). GA implementation and related functions (GA_InitialPopulation, GA_Elite, GA_CM, and GA_SRM) are discussed in Appendix B.

### 5.2. Checking the Fitness of Individuals: Check Function

This function is aimed at computing all the factors of the objective function of the problem for a candidate solution, i.e., the constraints of coverage and speech quality, and the value of the objective function. As shown in Algorithm 2, the input of the *Check* function is the 3D coordinates of *D* drones (X) and users’ location $\left\{w_{k}\;|\;k\in U\right\}$. Each drone in the candidate solution is assessed in three steps by three different functions. First, the collection of ground users associated to the AP is determined through the *Associate* function. Then, the QoS level of associated users is evaluated with the *QoS_eval* function. Finally, UAV endurance is determined (i.e., *Endurance_eval* function). Next, we elaborate on these functions.
**Algorithm 2:** Check function pseudocode.
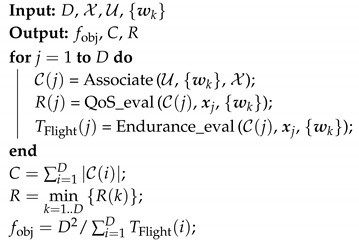


#### 5.2.1. Signal Coverage Evaluation (Associate Function)

This function associates each ground user to an aerial AP. A successful association requires that the Received Signal Strength Indicator (RSSI) and the Signal-to-Interference-plus-Noise-Ratio (SINR) are above a lower threshold. If multiple access points were available, each user will associate with the AP which provides the strongest RSSI. The output of the *Associate* function is the set of users that are associated to drone *j*, C(j):(11)$$\begin{array}{c} C\left(j\right)=\left\{i\in U\;|\;\gamma_{ik}=1,\;{\mathrm{RSSI}}_{ij}>{\mathrm{RSSI}}_{ik},\;\forall k\neq j\right\} \end{array}$$
where $\gamma_{ij}$ as a boolean that represents if user i∈U satisfies minimum thresholds (i.e., both, ${\mathrm{SINR}}_{ij}\geq {\mathrm{SINR}}_{\min}$ and ${\mathrm{RSSI}}_{ij}\geq {\mathrm{RSSI}}_{\min}$) when associated to UAV j∈D.

Observe that the user-to-AP association determines the Modulation and Coding Scheme (i.e., ${\mathrm{MCS}}^{\left(i\right)}$) at the physical layer of each WiFi station, which depends on the received ${\mathrm{RSSI}}^{\left(i\right)}=\max_{j=1\ldots D}\left\{{\mathrm{RSSI}}_{ij}\right\}$ and the required minimum sensitivity for each MCS. Next, we develop how RSSI and SINR are computed for each ground user.

##### UAV-to-Ground Channel Model

Most works [22,33,52] consider that ground users receive three different groups of signals that could be classified as: (a) Line-of-Sight (LoS), (b) Non-Line-of-Sight (NLoS), and (c) other reflected components which cause multipath fading. Each group has its own probability of occurrence depending on the environment, density, height, and elevation angle; so their impact on the overall path loss can be averaged. According to [53], the probability of fading is significantly lower than the probability of receiving the LoS ($P_{\mathrm{LoS}}$) and NLoS ($P_{\mathrm{NLoS}}$) components. Therefore, the UAV-to-Ground path loss can be represented by:(12)$$L\left(d,\theta \right)=P_{\mathrm{LoS}}\left(\theta \right)\;L_{\mathrm{LoS}}\left(d\right)+P_{\mathrm{NLoS}}\left(\theta \right)\;L_{\mathrm{NLoS}}\left(d\right)$$
which depends on the distance *d* between the UAV and the ground terminal, and θ that represents the elevation angle (in radians). $L_{\mathrm{LoS}}$ and $L_{\mathrm{NLoS}}$ represent the average path loss for LoS and NLoS components, respectively; and can be expressed as [52]:(13)$$\begin{array}{c} L_{\mathrm{LoS}}\left(d\right)=20\log_{10}\left(\frac{4\pi f_{c}\;d}{c}\right)+\xi_{\mathrm{LoS}} \end{array}$$(14)$$\begin{array}{c} L_{\mathrm{NLoS}}\left(d\right)=20\log_{10}\left(\frac{4\pi f_{c}\;d}{c}\right)+\xi_{\mathrm{NLoS}} \end{array}$$
where the first term accounts for the free space propagation loss, $\xi_{\mathrm{LoS}}$ and $\xi_{\mathrm{NLoS}}$ represent any additional losses due to the environment (a list of values can be found in [22]), and $f_{c}$ is the channel frequency.

Finally, the probability of receiving each component is given by [54]:(15)$$\begin{array}{c} P_{\mathrm{LoS}}\left(\theta \right)=\frac{1}{1+\alpha e^{-\beta \theta \cdot 180/\pi +\beta \alpha }} \end{array}$$(16)$$\begin{array}{c} P_{\mathrm{NLoS}}\left(\theta \right)=1-P_{\mathrm{LoS}}\left(\theta \right) \end{array}$$
where α and β are environment-dependent constants (e.g., rural, urban, etc.) as defined in [22] (e.g., α=β=0 for no buildings).

##### Obtaining RSSI and SINR

The Received Signal Strength Indicator (RSSI) represents the power received at the ground terminal. Let us consider a generic user *i* and the AP installed at drone *j*. Then, ${\mathrm{RSSI}}_{ij}$ can be calculated by subtracting the UAV-to-Ground path loss to the total transmitted power: (17)$$\begin{array}{c} {\mathrm{RSSI}}_{ij}=P_{\mathrm{tx}}+G_{ij}-L\left(∥w_{i}-x_{j}∥,\theta_{ij}\right) \end{array}$$(18)$$\begin{array}{c} G_{ij}=10\log_{10}\left(10^{G_{\max}/20}\cdot \cos^{2}\theta_{ij}\right) \end{array}$$
where $P_{\mathrm{tx}}$ (dBm) represents the power delivered by the transmitter antenna; $G_{ij}$ (dB) stands for the gain of the antenna between user *i* and drone *j* as indicated in Equation (Equation 18) (which is bounded by $G_{\max}$); L(d,θ) represents the path loss function introduced in Equation (Equation 12); and $∥w_{i}-x_{j}∥$ is the Euclidean distance between user *i* and drone *j*.

The Signal-to-Interference-plus-Noise-Ratio (${\mathrm{SINR}}_{ij}$) can be calculated by subtracting every noise and co-channel interferences to the received signal strength. To do so, let us declare $Z_{j}$ as the collection of drones that may cause interferences (channel planning is out of the scope of this paper) to the network deployed by UAV *j*. Then, the SINR (in dB) can be expressed as: (19)$$\begin{array}{c} {\mathrm{SINR}}_{ij}={\mathrm{RSSI}}_{ij}-F-N-\sum_{k\in Z_{j}}\left({\mathrm{RSSI}}_{ik}\right) \end{array}$$(20)$$\begin{array}{c} N=-174+10\log_{10}\left(C_{BW}\right) \end{array}$$
where *N* is the thermal noise, *F* accounts for the receiver’s noise figure, and $C_{BW}$ represents the signal bandwidth, which depends on the IEEE 802.11 standard revision under consideration.

#### 5.2.2. Quality of Service Evaluation (QoS_eval Function)

After establishing the association between users and APs (i.e., drones), which enables the verification of the coverage constraint, the second step consists of assessing the speech quality level (and hence, allow for the verification of the second constraint in Equation (Equation 9)) by computing the *R* factor (as introduced in Section 3). To do so, we evaluate the network performance on each set of users plus its corresponding UAV (i.e., C(j) + AP at drone *j*). This evaluation is based on a model of the IEEE 802.11 MAC sub-layer and provides delay and packet loss in the WiFi network, which allows one to calculate the *R* factor for each drone.

The IEEE 802.11 MAC sub-layer features a distributed access mechanism called DCF (i.e., Distributed Coordination Function), whose performance has been largely studied in literature [55,56,57,58]. In a nutshell, each contending station senses the medium during a period (DIFS) before transmitting in order to avoid collisions. If the channel is busy, the station waits for a random backoff interval before trying again (up to *M* attempts). The backoff procedure is also triggered between two consecutive transmissions. During the backoff process, a random number of discretized time slots ($T_{\mathrm{slot}}$) between 0 and $2^{i}W_{o}$ is selected, where $W_{o}$ accounts for the minimum contention window value, and *i* increases by one with each failed attempt up to reaching an upper bound *m*.

In this paper we have adapted some existing models of the DCF behaviour [59,60,61,62] to fit our context of heterogeneous traffic sources, non-saturated stations and a non-ideal channel. The central variable in DCF models is τ, which stands for the probability that an observed station attempts to transmit during a randomly chosen time slot. We use the expression proposed in [59]:(21)$$\tau =\frac{1}{\eta }\frac{1}{1-q}\left(\frac{r^{2}W_{o}}{\left(1-p\right)(1-{\left(1-r\right)}^{W_{o}}}-qr\left(1-p\right)\right)$$
(22)$$\begin{array}{cc} \eta =\left(1-r\right)+ & \frac{r^{2}W_{o}\left(W_{o}+1\right)}{2(1-{\left(1-r\right)}^{W_{o}})}+\frac{W_{o}+1}{2(1-q)}\left(\frac{r^{2}qW_{o}}{1-{\left(1-r\right)}^{W_{o}}}+rp\left(1-q\right)-rq{\left(1-p\right)}^{2}\right)\\ & +\frac{p}{2(1-q)(1-p)}\left(\frac{r^{2}W_{o}}{1-{\left(1-r\right)}^{W_{o}}}+qr{\left(1-p\right)}^{2}\right)\left(2W_{o}\frac{1-p-p{\left(2p\right)}^{m-1}}{1-2p}+1\right) \end{array}$$
where η is defined in Equation (Equation 22) and *p* is the probability that a packet suffers transmission errors due to collisions and/or noise. The VoIP traffic is modelled (we assume Poisson packet arrivals and the small queue assumption as discussed in [58]). through two variables, *r* and *q*, which account for the probability that at least one packet arrives during an idle state, and the probability that the MAC queue is not empty, respectively.
(23)$$\begin{array}{c} r=1-e^{-\lambda E[T]} \end{array}$$
(24)$$\begin{array}{c} q=1-e^{-\lambda E[T]E[B]} \end{array}$$
where E[B] is the expected number of back-off slots that a packet waits before transmission. This can be expressed as proposed in [62]:(25)$$E\left[B\right]=\frac{W_{o}}{2(1-p)}\left(\frac{1-p-{\left(2p\right)}^{m}}{\left(1-2p\right)}-2^{m}p^{M+1}\right),$$
and E[T] is the average slot duration, which represents the expected length of any state of the Markov chain. E[T] can be calculated by averaging the duration of each type of event with its probability (i.e., success, collision, corruption or idle) as proposed in [12,61].

Recall that each station in the system composed of drone *i* + C(i) (from now on termed set S) has it own τ since we consider heterogeneous sources. So let us tag each station as $i\in \left\{1,\ldots ,S\right\}$ and let the superscript (i) relate each variable to the *i-th* station. Then, $p^{\left(i\right)}$ is the probability that a packet from station *i* suffers any transmission error:(26)$$\begin{array}{c} p^{\left(i\right)}=\left(1-P_{i}^{\left(i\right)}\right)\cup {\mathrm{FER}}^{\left(i\right)}=1-P_{i}^{\left(i\right)}+P_{i}^{\left(i\right)}\cdot {\mathrm{FER}}^{\left(i\right)} \end{array}$$(27)$$\begin{array}{c} P_{i}^{\left(i\right)}=\prod_{j=1,\;i\neq j}^{\left|S\right|}\left(1-\tau^{\left(j\right)}\right) \end{array}$$
where $P_{i}^{\left(i\right)}$ represents the probability that only station *i* is attempting to transmit, and ${\mathrm{FER}}^{\left(i\right)}$ stands for the Frame Error Rate (in the access point case, MCS and FER are dynamically changed according to its communication partner. Our approach is to average its data bit-rate and FER) (i.e., channel noise), which can be obtained as:(28)$${\mathrm{FER}}^{\left(i\right)}=1-{\left(1-P_{e}^{\left(i\right)}\right)}^{L}$$
where *L* is the packet size (considering preamble, headers and payload), and $P_{e}^{\left(i\right)}$ represents the bit error rate of station *i*, which can calculated by considering the user’s ${\mathrm{MCS}}^{\left(i\right)}$ as proposed in [63,64].

Finally, one can solve the non-linear equation system formed by Equations (Equation 21) and (Equation 26). Since the downlink performance is the dominant factor in QoS as demonstrated in [13], the packet loss and delay (the small queue assumption allows us to neglect the queuing delay and DEL can be approximated by the channel access delay) at the access point (tagged as i=AP) will represent the worst case. As such, it will be used to calculate the QoS for the set S. Then, the packet loss ratio and delay can be expressed as: (29)$$\begin{array}{c} \mathrm{PL}=1-\frac{\left(1-{\mathrm{FER}}^{\left(\mathrm{AP}\right)}\right)\tau^{\left(\mathrm{AP}\right)}\prod_{j=1,j\neq \mathrm{AP}}^{\left|S\right|}\left(1-\tau^{\left(j\right)}\right)}{\lambda^{\left(\mathrm{AP}\right)}\;E\left[T\right]} \end{array}$$(30)$$\begin{array}{c} \mathrm{DEL}=E\left[B^{\left(\mathrm{AP}\right)}\right]\cdot E\left[T\right] \end{array}$$

Substituting PL and (DEL+20) (recall that the delay includes the a codec packetization delay of 20 ms) in the terms $P_{pl}$ and *d* from Equation (Equation 1) provides the output of the *QoS_Eval* function: the *R* factor.

#### 5.2.3. Endurance Evaluation (Endurance_eval function)

This function estimates the expected UAV’s endurance. As introduced in Section 3, the flight duration has two main components: the power (in Watts) required to hover the drone ($P_{\mathrm{Hover}}$), and the power consumption at both radio links (i.e., recall from Section 3.2 that it was proportional to $P_{\mathrm{WiFi}}$).

$P_{\mathrm{Hover}}$ in rotor-based aircrafts can be easily derived from the Momentum Theory [29] as:(31)$$P_{\mathrm{Hover}}=\frac{g^{\frac{3}{2}}}{\eta_{ps}r_{p}\sqrt{2N_{R}\rho_{a}\pi }}m^{\frac{3}{2}}$$
where *g* accounts for gravity acceleration, *m* represents the mass of the aircraft, $\eta_{ps}$ is the rotorcraft propulsion system efficiency, $N_{R}$ is the number of rotors, $r_{p}$ represents the radius of each rotor and finally $\rho_{a}$ represents the density of air. $N_{R}$, $r_{p}$ and $\eta_{ps}$ are UAV-specific. While the first two are fixed and can be readily obtained from manufacturer manuals, the latest is variable (although a constant value is generally adopted) and requires experimental estimation [48].

Regarding $P_{\mathrm{WiFi}}$ at the access point, the power can be expressed as:(32)$$\begin{array}{cc} P_{\mathrm{WiFi}} & =\frac{1}{E[T]}\cdot (\underset{\mathrm{idle}\;\mathrm{interval}}{\underbrace{J_{\sigma }\;P_{i}}}\\ \; & +\underset{\mathrm{transmission}\;\mathrm{interval}}{\underbrace{J_{\mathrm{tx},s}\;P_{\mathrm{tx},s}+J_{\mathrm{tx},e}\;P_{\mathrm{tx},e}+J_{\mathrm{tx},c}\;P_{\mathrm{tx},c}}}\\ \; & +\underset{\mathrm{reception}\;\mathrm{interval}}{\underbrace{J_{\mathrm{rx},s}\;P_{\mathrm{rx},s}+J_{\mathrm{rx},e}\;P_{\mathrm{rx},e}+J_{\mathrm{rx},c}\;P_{\mathrm{rx},c}}}) \end{array}$$
where $J_{\sigma }$ represents the energy spent during an idle interval; $J_{\mathrm{tx},s}$, $J_{\mathrm{tx},e}$, and $J_{\mathrm{tx},c}$ represent the average energy consumption (in Joules) during a successful, erroneous or collided transmission at the access point respectively; and $J_{\mathrm{rx},s}$, $J_{\mathrm{rx},e}$ and $J_{\mathrm{rx},c}$ represent the expected energy consumption during a successful, erroneous or collided reception respectively. The terms $P_{i}$, $P_{\mathrm{tx},s}$, $P_{\mathrm{tx},e}$, $P_{\mathrm{tx},c}$, $P_{\mathrm{rx},s}$, $P_{\mathrm{rx},e}$ and $P_{\mathrm{rx},c}$ indicate the probability of such events. A detailed expression of the factors in (Equation 32) based on the traffic and our model of the mac-sublayer is deduced in Appendix A.

Finally, the expected flight duration $T_{\mathrm{Flight}}$ of a particular drone can be calculated by substituting the expressions of $P_{\mathrm{Flight}}$ and $P_{\mathrm{WiFi}}$ in Equation (Equation 7).

### 5.3. Example Solution

This section offers a first example solution and validates the models used in the previous section with the network-simulator ns-3. We define a scenario where 40 ground users are randomly placed in a square area of (100 m × 100 m). The parameters used in the scenario are listed in Table 3. Unless otherwise specified, such settings are common to all experiments in this paper.

The solution to the optimization problem has been found by implementing Algorithm 1 in Matlab^®^. A graphical representation of the solution is provided in Figure 4. Notice that different colors represent different WiFi associations between users and UAVs.

The scenario (including the UAVs) has been simulated with the ns-3 network simulator. In the simulation, no interferences between APs have been considered and transmission buffer sizes have been set to one packet to match the small queue assumption made in Section 5.2.2.

The results obtained are summarized in Table 4, which shows the most relevant performance measurements (location, flight time, quality level and radio power). We have performed 30 runs of simulations and the results presented represent average values.

The results in Table 4 indicate that the solution meets both coverage and speech quality constraints, and the QoS and energy models used in Algorithm 1 are aligned with simulation results, which validates the models proposed in Section 5.2.2.

## 6. Solving a Dynamic Scenario

So far, we have applied the GA-based search algorithm in Section 5 for solving the problem of the initial deployment of drones to provide service to a set of users who are assumed to be static at known positions. However, in some activities (e.g., outdoor concerts), users are free to move within a bounded area. Consequently, the problem constraints may not hold after a while due to users’s movement unless deployed UAVs are continuously relocated. One approach to do this is to perform periodic updates of the user’s positions every $T=T_{\mathrm{sample}}$ seconds, (re)solve the optimization problem, and update UAVs locations, as illustrated in Figure 5. A central assumption in this approach (and our work hypothesis) is that the set of UAVs initially deployed and their position will suffer small changes (assuming that the trajectories always minimize UAVs displacement) if the sampling period is short enough.

Between two consecutive relocation periods *T*, one has to consider the relocation delay, (i.e., drones updated their location), that includes:The time required to solve the problem, $T_{\mathrm{solve}}$, which in turn depends on the performance of the algorithm used to find the solution to the problem.The time required to relocate UAVs, $T_{\mathrm{relocate}}$, which depends on the distance traveled by UAVs and their speed. Small UAVs typically travel at speeds below 15 m/s [30].

Under the assumption that AUVs will complete self relocation quickly due to high speed and short distance, the main limiting factor for a fast sampling of user positions is the time required to solve the problem. For this reason, we want to reduce the execution time of the search algorithm for periodic relocation.

According to [65], search algorithms based on Particle Swarm Optimization (PSO) tend to be computationally more efficient than GA-based algorithms due to faster converge. However, although both metaheuristics provide similar effectiveness (i.e., solution quality), it is also known that PSO is more prone than GA to get stuck on local solutions. For this reason, we propose to use the GA-search algorithm from Section 5 only for the initial deployment, but then use a PSO-based algorithm to solve the UAVs placement for the remainder of the mission. PSO algorithms have been successfully used to solve optimal 3D placement of UAVs before [43].

### 6.1. Periodical Search Algorithm: PSO

Particle Swarm Optimization (PSO) is a heuristic search method inspired by bird flocking or fish schooling’s social behaviour. In PSO, each candidate solution is represented as a particle that moves around in the search-space towards better solutions. Initially, a collection of particles with random positions in the search-space is created. Each particle is represented by its position (i.e., collection of characteristics, similar to individuals in GA), and a velocity. On each iteration, each particle updates its position based on its velocity, its best known position, and the global best location.
**Algorithm 3:** PSO-based search pseudocode.
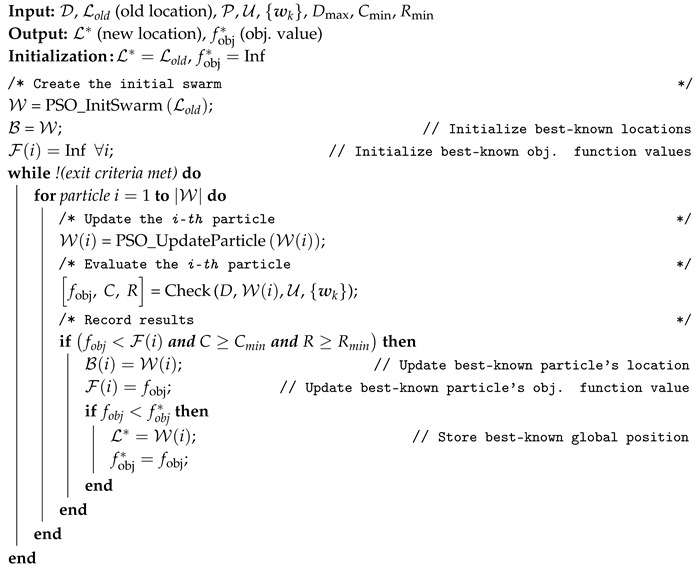


Algorithm 3 shows the pseudocode for our PSO-based search algorithm. The input of the algorithm is the previous positioning (D and $L_{\mathrm{old}}$), the set of ground users (U) and their location ($\left\{w_{k}\right\}$), the set of edges (P), coverage ($C_{\min}$) and quality ($R_{\min}$) constraints, and the maximum number of drones ($D_{\max}$). The algorithm assumes that the number of deployed drones (*D*) does not change between two consecutive executions, and relocates them. To do so, it initializes the swarm of particles (W) with the function *PSO_InitSwarm* around the previous positioning ($L_{\mathrm{old}}$). Then, each particle is updated through the function *PSO_UpdateParticle*, which receives as input the best known position for that particle. Then, the candidate position (X) is evaluated through the function *Check* from Section 5. Finally, if the particle improves its result, we store the new position as W(i) and, if the best solution is improved, we record it at $L^{*}$, which is the output of the algorithm. This procedure is repeated until the exit criteria is met. PSO implementation and its functions (*PSO_InitSwarm* and *PSO_UpdateParticle*) is discussed in Appendix C.

### 6.2. Convergence Speed

In this section, we compare the convergence speed of our two heuristics (GA and PSO) to validate PSO’s faster convergence speed and justify its use for periodical relocation of UAVs.

We have performed the following experiment. We have placed 100 users on different sized areas (from 1000 to 10,000 m^2^). Ground users move according to a correlated random walk model (the rest of the settings are shown in Table 3). Each user moves at a walking speed of 5.3 km/h with a probability of 0.8, and in such case, the direction remains unchanged with probability 0.8. Users rotate 180 degrees when they reach the area bounds. Algorithm 1 is used for the initial deployment, and the number of drones deployed *D* remains unchanged for the remainder of the simulation (15 min). To achieve this, we limit subsequent solutions to the best solution found with *D* drones (even though problem constraints were not satisfied). Every 30 s (T=30), we update drones’ location by executing one of the resolution methods: (a) GA, or (b) the PSO-based search we introduced in this section. Figure 6 illustrates the obtained results (x-axis represents the sparsity of users in m^2^/U). With every new solution, drones are relocated (considering a speed of 60 Km/h) so that the sum of the distances traveled by all drones is the minimum possible.

Results prove that using the PSO-based algorithm significantly improves the convergence speed, reporting up to an 80% improvement in the resolution time. With the PSO-based search, the resolution time ($T_{\mathrm{solve}}$) never takes longer than 25 s, which validates its use with our preferential sampling time (T=30 s). Results also show that scenarios with high user sparsity are easier to solve, and the algorithm takes less time to converge to solutions.

## 7. Numerical Results

This section provides a numerical analysis of the results obtained with our optimal placement method for two different VoWiFi communication services.

Individual VoIP channels. Each ground user has an independent bidirectional VoIP channel.Broadcast channel. Every ground user receives a single shared unidirectional VoIP flow.

For each communication service, we solve our optimal 3D placement problem for an increasing number of users confined in a small area of 100 m^2^ as well as in a large area of 10,000 m^2^. The results provided are the average of 30 simulation runs and exhibit a 95% confidence interval. On each run, users’ location was randomly generated.

### 7.1. Individual VoIP Channels

In this experiment, each user has an individual bidirectional VoIP channel. We vary the number of ground users from 10 to 100 over two terrain sizes (100 m^2^ and 10,000 m^2^) using the parameters listed in Table 3. We want to study the evolution of $f_{\mathrm{obj}}$ (i.e., the average rate of drones per hour of service), the power spent on communications $P_{\mathrm{Radio}}$ and the total required power (i.e., radio communication and hovering).

Figure 7b illustrates the drone turnover rate ($f_{\mathrm{obj}}$) in UAVs/hour. As introduced in Section 4, this value is the ratio between the number of drones *D* and the average flight duration ${\bar{T}}_{\mathrm{Flight}}$. Results suggest that scenarios with large user sparsity (m^2^/U) tend to demand more of drones. For example, from 10 to 50 users, the 10,000 m^2^ scenario requires more drones to satisfy the coverage constraint ($C\geq C_{\min}\cdot U$) than that of 100 m^2^. The average drone flight duration was around 25 min.

Figure 7b shows the average power spent in communications (i.e., $P_{\mathrm{Radio}}$). In this graph, we can identify two behaviours:For less than 50 users, drones spend more radio power in the smaller scenario than in the larger one. This can be explained as follows: while in the bigger scenario up to $\left(1-C_{\min}\right)\cdot U$ users can be excluded from coverage by locating drones far away from them, the smaller one has not enough terrain for the drones to exclude people (C=U), which in turn increases the average number of users associated to each aerial AP.For more than 50 users, the previous behaviour gets inverted when the user density is high enough. This is attributable to solutions where drones are placed at higher altitudes. In general, drones increase their altitude to cover a larger area (i.e., antenna directivity), which in turn reduces the received signal strength and the achievable data bit-rate (i.e., a slower MCS is selected), so the time to complete data transmission is longer which translates into higher energy consumption.

Finally, Figure 7c represents the total power used by all UAVs broken down into mechanical power (i.e., $P_{\mathrm{Flight}}$) and communications power (i.e., $P_{\mathrm{Radio}}$). In our experiment, $P_{\mathrm{WiFi}}$ represents between the 14% and 16% of the total power expenditure, scaling with user density. This ratio is highly dependent on drone characteristics and the power consumption of the radio network interfaces. A careful selection of these aspects is a key factor for reducing the number of required UAVs per hour of service.

### 7.2. Broadcast Channel

Our second experiment considers a single broadcast VoWiFi channel. This is a one-way communication service could be used for announcements to ground users. To compare both VoWiFi communication services, we set up the same range of users and terrain sizes as in our previous experiment (and continue using the settings listed in Table 3). As a particularity, broadcast channels are transmitted using the lowest available bit-rate modulation (i.e., MCS 0 in IEEE 802.11n) to maximize coverage.

Figure 8 shows the results obtained for $f_{\mathrm{obj}}$, $P_{\mathrm{WiFi}}$ and the total power consumption. The average drone turnover rate ($f_{\mathrm{obj}}$) shown in Figure 8a depends on the size of the terrain, but the number of served users does not affect negatively since the problem constraint related to QoS is no longer an obstacle due to the scarce traffic load (i.e., there is only one traffic flow from the AP to all its associated users).

Figure 8b shows that the power spent in radio communications is close to the power consumption of the radio idle state ($P_{\mathrm{Radio}}\approx 2\cdot \rho_{\mathrm{idle}}$). This shows that the channel occupation is very low. Indeed, the codec G.711 requires a throughput of 64 Kbps (around 80 Kbps at the physical layer), and the available data bit-rate is 6.5 Mbps. As a result, $P_{\mathrm{Radio}}$ represents a 14% of the total power (see Figure 8c).

Finally, let us compare the total required power in both experiments (see Figure 7c and Figure 8c). Since the VoWiFi broadcast service require fewer drones than the individual, the total power consumption is also improved. Energy savings are reduced up to 20% on the largest scenario and up to a 75% on the small one. As expected, energy savings increase with user density and thus, QoS-constrained scenarios take advantage of replacing individual VoIP channels with a unique broadcast channel.

### 7.3. Comparing with Other Approaches

As shown in Section 2, several research works have addressed UAVs’ optimal 3D deployment for communications. Unfortunately, comparing our results with these works is very difficult due to: (a) the objective function and constraints are different; (b) the radio access network (WLAN versus cellular) is also different, which affects coverage and QoS. But for the sake of completeness, we compare our results with the following approaches:A search method based on K-means clustering that seeks the minimum number of UAVs that satisfies the optimization problem in this paper, Equation (Equation 9). K-means clustering has also been used for 3D optimal placement in [41].Our previous work [12] for optimal 3D placement in search and rescue missions. This GA-based search algorithm is the only related work that has comparable constraints (i.e., speech quality in WiFi) although the objective function is not the same and does not model UAVs endurance (see Table 1 for details).

For the comparison, we vary the number of ground users from 10 to 100 over a 10,000 m^2^ terrain, using the rest of parameters we listed in Table 3 (some parameters, such as coverage ratio, are not supported by our previous work [12]). Then, we analyze the evolution of the drone turnover rate ($f_{\mathrm{obj}}$) and the total power expenditure (i.e., adding the power of all UAVs in the solution).

Figure 9 illustrates how the UAV turnover rate (left axis) and the total power expenditure (right axis) change while the number of ground users increases. As expected, both variables are interrelated, and scale up with the number of ground users. Results suggest that our new proposal can reduce both, UAVs turnover rate and energy consumption. Compared to the K-means search method, our method saves 60% of the energy consumption for U=10, and 80% for U=100. Compared to [12], our method also provides energy savings, although this is reduced up to 22%. In general, our method also reduces the number of UAVs per hour required to complete the mission (e.g., reduced from 9 to 7 in U=70).

## 8. Conclusions and Future Research

This paper formulates a new problem for the UAVs’ optimal 3D placement to offer VoWiFi services with guaranteed speech quality to ground users confined in a known area. Our objective function minimizes the ratio between the number of UAVs initially deployed and their average endurance expectation, providing the layout that requires fewer drones per hour of service. Solutions are found using two well-known search methods: (a) a genetic algorithm is proposed for the initial deployment, and (b) a particle swarm optimization algorithm is proposed for updating UAVs’ location in scenarios with users mobility.

A numerical analysis compares the results of our optimal placement algorithm for unicast and broadcast VoWiFi services. Our results for the unicast service show that the drone turnover rate is related to both, users’ sparsity and the number of associated users to each drone. In terms of energy consumption, radio communications represent between a small part (between 14% and 16%) of the total power consumption depending on the density of users served per drone. Large areas tend to locate drones at higher altitudes, which increases UAVs’ mechanical energy consumption. The broadcast VoWiFi service generates a very low traffic load, which translates into a significant reduction of UAVs deployed and total power consumption (between 20% and 75%), and less sensitivity to the number of served users.

The next step in our work is to consider two new aspects in this field: (a) interferences between APs, so that overlapping channels could negatively impact on QoS; and (b) using 5G instead of WiFi as the enabling technology for the UAV-to-Ground access network.

## Figures and Tables

**Figure 1 sensors-20-04455-f001:**
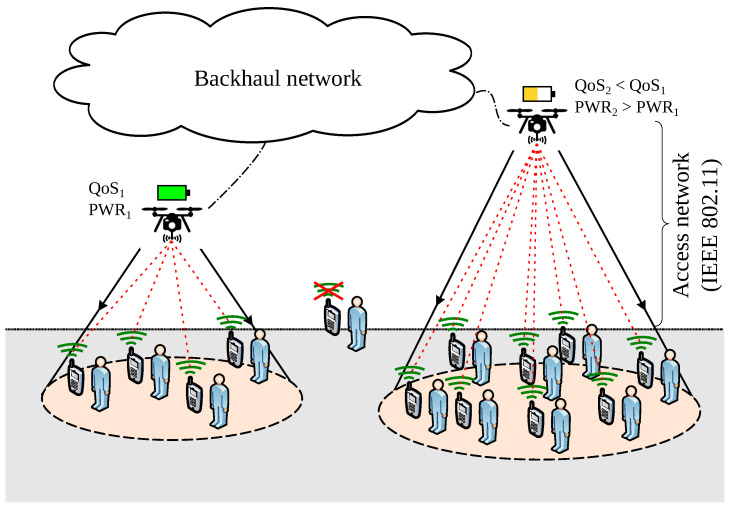
UAV-based VoWiFi network.

**Figure 2 sensors-20-04455-f002:**
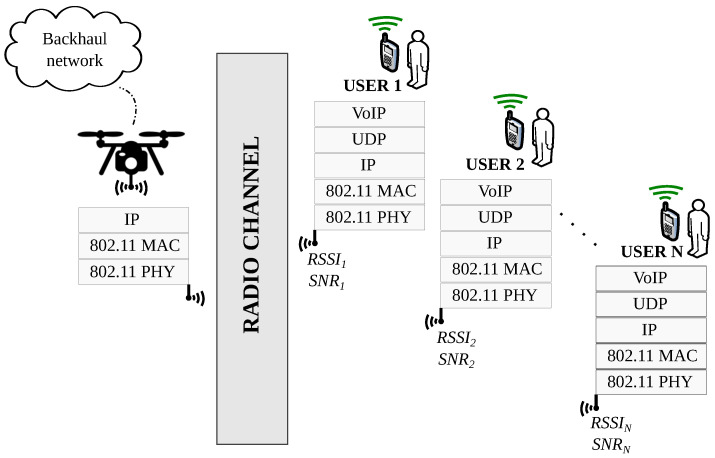
Communication system for each drone.

**Figure 3 sensors-20-04455-f003:**
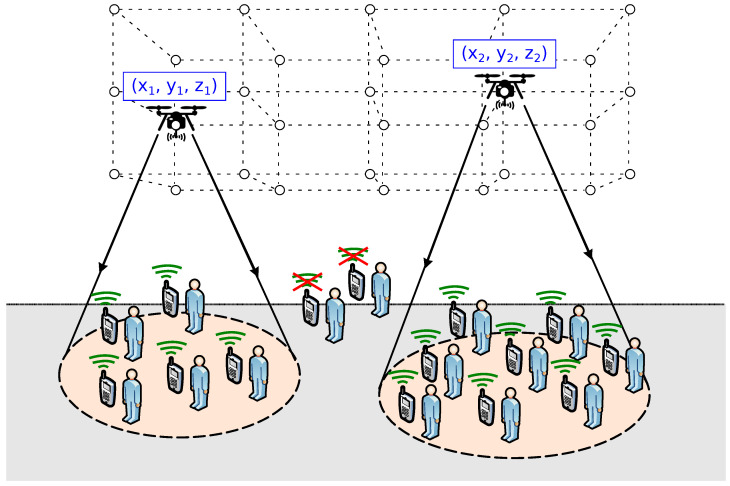
UAVs placement scenario.

**Figure 4 sensors-20-04455-f004:**
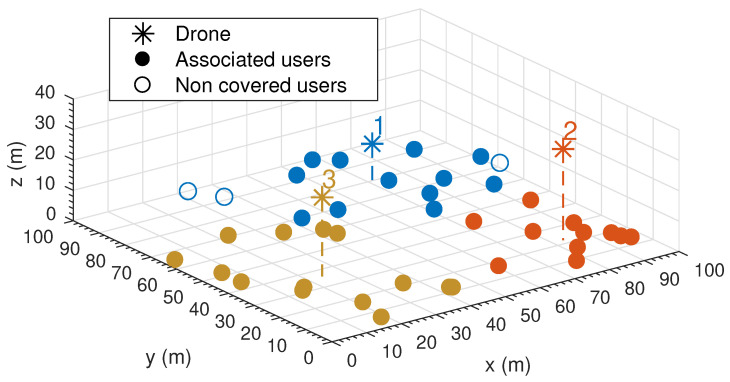
Example solution.

**Figure 5 sensors-20-04455-f005:**
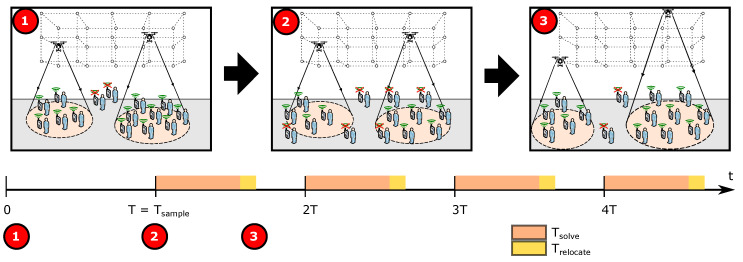
Dynamic scenario.

**Figure 6 sensors-20-04455-f006:**
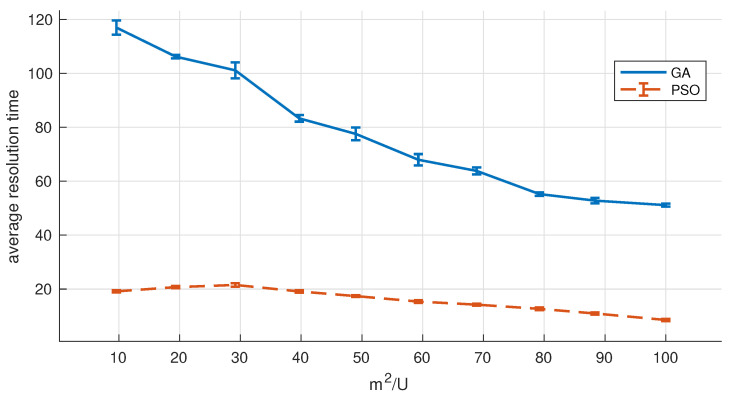
Resolution time ($T_{\mathrm{solve}}$).

**Figure 7 sensors-20-04455-f007:**
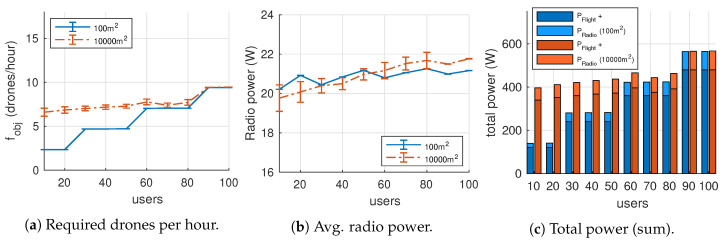
Results for the individual VoIP channels experiment.

**Figure 8 sensors-20-04455-f008:**
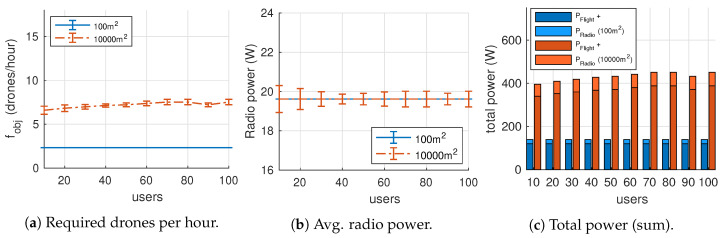
Results for the broadcast VoIP channels experiment.

**Figure 9 sensors-20-04455-f009:**
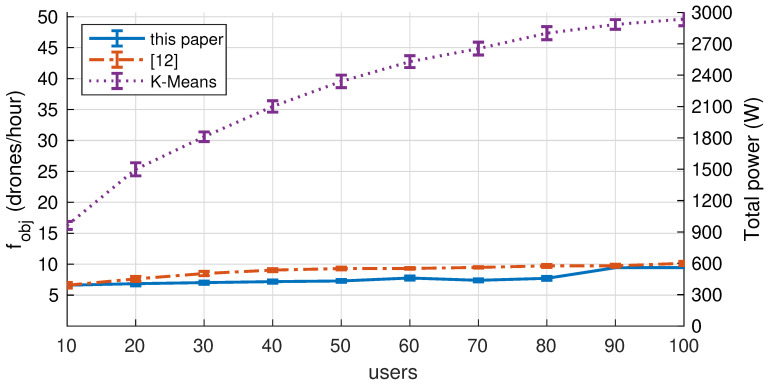
Comparison against other approaches.

**Table 1 sensors-20-04455-t001:** Comparison of main works related to 3D optimal placement of UAVs for communication.

Paper	Objective	Multi-UAV	UAV-to-Ground Technology	UAV Mechanical Power	UAV Radio Power	User Radio Power	Speech Quality Constraint
[26]	minimize transmit power	no	cellular	no	yes	no	no
[25]	maximize coverage with minimum transmit power	no	cellular	no	yes	no	no
[31]	maximize capacity and extend battery life of mobile users	yes	cellular	no	yes	yes	no
[32]	minimize transmit power of mobile users (deployment and mobility)	yes	cellular	no	no	yes	no
[22]	minimize number of UAVs	yes	cellular	no	no	no	no
[39]	minimize the total transmit power of the entire aerial system	yes	cellular	no	yes	no	no
[42]	minimize total power consumption given predicted traffic	yes	cellular	yes	yes	no	no
[43]	minimize total power consumption	yes	cognitive radio	yes	yes	no	no
[12]	minimize number of UAVs	yes	WiFi	no	no	yes	yes
this paper	minimize the ratio between the number of UAVs and UAVs’ endurance	yes	WiFi	yes	yes	no	yes

**Table 2 sensors-20-04455-t002:** 802.11 sensitivity relations for OFDM modulations with 20 MHz channels.

Modulation	Coding Rate	Data Rate (Mb/s)	Sensitivity (dBm)
BPSK	1/2	6	−82
BPSK	3/4	9	−81
QPSK	1/2	12	−79
QPSK	3/4	18	−77
16-QAM	1/2	24	−74
16-QAM	3/4	36	−70
64-QAM	2/3	48	−66
64-QAM	3/4	54	−65

**Table 3 sensors-20-04455-t003:** Example solution input parameters.

IEEE Standard		Scenario		Traffic		Constraints		Energy	
Revision	802.11 n	Users	40	Calls/hour/user	5	${\mathrm{RSSI}}_{\min}$	−82 dBm	$E_{\mathrm{Battery}}$	60 Wh
GI	800 ns	Size	100 m × 100 m	Call length	180 s	${\mathrm{SNR}}_{\min}$	20 dB	$P_{\mathrm{Hover}}$	120 W
Preamble	Greenfield	X-Y-Z step	1 m	VoIP codec	G.711	$R_{\min}$	65	$\rho_{\mathrm{tx}}$	16 W
Bandwidth	20 MHz	Altitude layers	{10, …, 40} m	On/Off times	CBR	$C_{\min}$	0.9	$\rho_{\mathrm{rx}}$	9.7 W
Retries	7	Prop. Exponent	3.3	Packet interval	20 ms			$\rho_{\mathrm{idle}}$	9.7 W

**Table 4 sensors-20-04455-t004:** Algorithmic and simulation results.

Drone	$x_{i}$	|C(i)|	$T_{\mathrm{Flight}}$ (min)	Algorithm 1/Simulation	
				*R*	$P_{\mathrm{Radio}}$ (W)
1	(69,77,14)	12	25.61	89 / 88	20.56 / 20.27
2	(83,22,30)	11	25.64	90 / 89	20.42 / 20.16
3	(24,36,26)	14	25.56	87 / 85	20.86 / 20.53

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
