# Peer review of "Energy-Efficient UAVs Deployment for QoS-Guaranteed VoWiFi Service"

_sensors, 2020, doi:10.3390/s20164455_

Round 1

Reviewer 1 Report

Dear author,

THere are many paragraphs that need to rewrite to avoid plagiarism. please check the comments in the paper.

Equation 3, 4, 5, 6 need to be mentioned in the paper

The paper did not follow the journal's format.

There is no acknowledgment.

There is no Declare for Conflict of Interest before References.

At the end of the References, authors need to check the format again.

Reviewer 2 Report

The paper proposes a new optimization problem that finds the number and position of 67 UAVs to be deployed to cover an user-defined portion of ground users that provides the minimum 68 drone turnover rate with guaranteed VoIP QoS.

In the eq.1 to calculate the R value, H is defined as the Heaviside function as H(x). What is x representing here?

The paper presents an interesting problem of VoIP on a UAV. The authors analysis shows that adding an AP adds 15% to the power draw. The paper is detailed and well written

Reviewer 3 Report

This work presents a methodology to deploy UAVs to provide Access Points (AP) for a number of users, aiming at reducing the turnover, defined by the authors as the number of UAVs to be deployed divided by the number of hours to provide the service. 

The authors propose to address the problem as an optimisation problem using Genetic Programming and Particle Swarm Optimisation techniques to reach a solution. 

My concerns with this work is related to the assumption and test scenarios. 

I am aware that it is difficult to model all the possible scenarios and variables such as number of users, UAV's height, energy consumption, etc. The authors for instance, model a dynamic scenario where users are added to the communication area or some of them leave. 

However, it is not clear how effective the authors method is. From figures 7 and 8, the turn over does not seem to get reduced. 

I understand that part of the goal is find the number of UAVs needed to provide coverage in terms of the number of users, Quality of Service, etc. However, for a more realistic scenario, noise in the communication, noise in the UAVs' position, time to be deployed, heights, should be considered and discussed. 

On the other hand. It is also difficult to appreciate the contribution of the work if no other implementation is presented, from other related work. 

For dynamic environments, it would be interesting to define the conditions for which a dynamic demand of the service becomes a problem. If users leave or arrive at the service zone, how does that affect the effectiveness of the UAVs to provide the service. If the number of users gets reduced significatively, do some of the UAVs can withdraw from the network? If the number of users increase dramatically, more UAVs have to be added, how does that change the optimisation function? 

Some amendments to be done:

Define QoS in the abstract.

Figure 8 has to be fixed since the captions for a) and b) overlapped with subfigure c).

Reviewer 4 Report

I find the paper very well written and clear, using concepts from different areas in an adequate way. I just have some concerns regarding the dynamic scenario. The presented approach does not take into account, apparently, the power consumption related to moving drones around to enable continuous coverage, which might be important. Also, the power costs and QoS related to user roaming seems not to be modeled there.

Finally, in equation (8) you use D^2 /Sum(T_i) as the cost function, while the previous descriptions seems to ask for the use of D/Sum(T_i). Why are you using the first function instead of the second? Could you clarify this cost function a little more? 

Also, why do you constraint your search to D*+1, maybe by increasing it further you may reduce your cost function. Please clarify.

Reviewer 5 Report

- The state of the art is not well-established. There are several articles which talk about UAV-ground energy requirements. An exhaustive search should be helpful.

- In motivation, the authors talk about their previous work. It is required to give the exact need for this solution, not research left out in the earlier works.

- There are articles which talk about deployment cost with better convergence. The authors need to show how their work is exactly different not just a comparison on parameters. The advantages must be shown in terms of gains (in comparison).

- The backhaul links need to be clarified. What exactly here the authors are referring to? How the backhaul is modelled and what role UAVs play there?

- There are some works to minimize the number of UAVs to cover the entire area. It is recommended to further explore the existing solutions.

- Optimization in (8) is static considering the initial model and assumptions, thus, what this optimization helps to attain?

- What is the advantages of this work in comparison with the other works which also use GA?

Round 2

Reviewer 1 Report

Dear authors,

there are many mistakes and you did not follow the journal's format. Please check the attachment file.

also need to check english grammar.

Author Response

We thank the reviewer for his/her comments.

We apologise for failing to realise that the reviewer wrote a generous amount of comments in the annotated pdf file, and thank the reviewer for his/her detailed comments. Most comments have been addressed to the best of our ability in the revised version. However, we were not able to handle the repeated comments regarding line numbers or citation format since these aspects are automatically arranged by the LaTeX engine and the journal’s LaTeX template (we used \documentclass[sensors,article,submit,moreauthors,pdftex]{Definitions/mdpi} in our latex document). As such, we believe that these aspects can be only fixed during the proof editing phase. We will make sure that proof editing accommodates these comments.

Regarding to English, we have thoroughly checked the grammar and corrected numerous errors. We have used the online service Grammarly for this task (besides manual inspection).

We also attach a document that shows the changes we made in this revision.

Reviewer 3 Report

I appreciate the time the authors have invested to address my observations. 

Yet, I find the problem assumptions somewhat artificial. As an optimisation problem I can accept the way the authors have  modelled and address the problem. From that point of view, the solution may be of interest to certain community. 

Author Response

We thank the reviewer for his/her comments. In this revision we have performed an in-depth revision of grammar and punctuation signs, which resulted into more than 200 minor edits.

We also attach a document that shows the changes we made in this revision.

Reviewer 5 Report

No further comments.

Author Response

(The authors gave the same response as above.)
